# Relationship between the Tokyo Guidelines and Pathological Severity in Acute Cholecystitis

**DOI:** 10.3390/jpm13091335

**Published:** 2023-08-30

**Authors:** Tae Young Park, Jae Hyuk Do, Hyoung-Chul Oh, Yoo Shin Choi, Seung Eun Lee, Hyun Kang, Soon Auck Hong

**Affiliations:** 1Division of Gastroenterology, Chung-Ang University Gwangmyeong Hospital, Gwangmyeong 14353, Republic of Korea; 2Division of Gastroenterology, Chung-Ang University Hospital, Seoul 06973, Republic of Korea; 3Department of Surgery, Chung-Ang University Hospital, Seoul 06973, Republic of Korea; 4Department of Anesthesiology and Pain Medicine, Chung-Ang University Hospital, Seoul 06973, Republic of Korea; 5Department of Pathology, Chung-Ang University Hospital, Seoul 06973, Republic of Korea; hsu108@cau.ac.kr

**Keywords:** acute cholecystitis, severity, Tokyo Guidelines, gallbladder, inflammation

## Abstract

**Background:** It is not well understood whether the severity of acute cholecystitis (AC) correlates with the extent of gallbladder (GB) inflammation or laboratory findings. This study aimed to assess whether the severity of AC, in accordance with the Tokyo Guidelines (TGs), is consistent with the extent of GB inflammation on histopathological and laboratory findings, including microbiological isolation in blood and bile. **Methods:** The medical records of patients who underwent cholecystectomy for AC between January 2017 and May 2020 were reviewed. Demographic data, laboratory findings, the microbiologic culture of blood and bile, the extent of GB inflammation, and stone composition were compared in accordance with the TGs. **Results:** A total of 217 patients were divided into three groups of increasing severity—Grade I (n = 146), Grade II (n = 51), and Grade III (n = 20)—in accordance with the TGs. The Grade III group contained significantly older patients compared with the Grade I or Grade II groups (Grade I, 56.9 ± 13.9; Grade II, 64.3 ± 15.4; Grade III, 69.9 ± 9.9; *p*-value < 0.001). Patients in the Grade III group showed significantly higher levels of CRP, WBC, creatinine, and bilirubin and lower levels of platelets and albumin compared with the Grade I or Grade II group. As the grade of severity increased, the rate of microbiological isolation in blood (Grade I, 0% [0/146]; Grade II, 2.0% [1/51]; Grade III, 20% [4/20]; *p*-value < 0.001) and bile (Grade I, 19.9% [29/146]; Grade II, 33.3% [17/51]; Grade III, 70% [14/20]; *p*-value < 0.001) also increased significantly. However, there were no significant differences in the extent of GB inflammation between grades. **Conclusions:** AC severity, as stated by the TGs, does not correlate with the extent of GB inflammation on histopathological and laboratory findings. However, microbiological isolation in blood and bile was increased proportionally to the grade of the TGs.

## 1. Introduction

Acute cholecystitis (AC) is a common gallstone-induced complication and is known to occur in 6–11% of patients with gallstones that were monitored for 7–11 years [1]. The disease has varying degrees of severity and can cause inflammation of the gallbladder (GB) wall, local complications, and, in particularly severe cases, multi-organ failure. AC can be diagnosed with at least one of the following inflammation symptoms: Murphy’s sign or abdominal pain, accompanied by fever, and an elevated C-reactive protein (CRP) level or white blood cell (WBC) count. Diagnosis is also possible if there is a characteristic imaging sign in clinically suspected cases [2].

The main treatment for AC is a laparoscopic cholecystectomy, preferably within 72 h of diagnosis [3]. However, depending on the patient’s surgical risk, the severity of GB inflammation, and the presence of other complications, other treatments such as GB drainage may be performed prior to surgery [4]. Therefore, the treatment should be determined by evaluating the patient’s surgical risk and the severity of GB inflammation along with conservative treatment [5]. Over the years, it has been claimed that AC can be conservatively treated in most cases, consequently delaying selective cholecystectomy. However, evidence accumulated in recent years has shown that early cholecystectomy for AC during acute hospitalization is safe and cost-effective. Specifically, data collected from centers with extensive experience in laparoscopic surgery suggest that early laparoscopic cholecystectomy may be an attractive, feasible, and safe treatment strategy.

The Tokyo Guidelines (TGs) are a global index used to determine the severity of AC [2]. However, whether the AC grade correlates with the extent of GB inflammation on histopathological or laboratory findings is not well understood at present. Therefore, this study aims to determine whether the TG grading of AC cases is consistent with the extent of GB inflammation found in the histopathological and laboratory findings, including microbiological isolation in blood and bile.

## 2. Methods

We reviewed the medical records of patients who underwent cholecystectomy for AC between January 2017 and May 2020 at Chung-Ang University Hospital. The study was also approved by the Institutional Review Board at Chung-Ang University Gwangmyeong Hospital (IRB No. 2302-059-009).

### 2.1. Patients

Inclusion criteria were (1) patients who had cholecystectomy for AC, and (2) age > 20 years. Exclusion criteria were (1) cholecystectomy without inflammation, such as for a GB polypoid lesion, adenomyomatosis, and GB cancer, (2) chronic cholecystitis combined with intrahepatic and/or common bile duct stones, (3) age < 19 years old, and (4) cholecystectomy due to biliary acute pancreatitis.

Demographic data (age, sex), laboratory findings (CRP, WBC, PLT, Cr, Alb, Bilirubin, AST, ALT, amylase, lipase, PT-INR), the microbiologic culture of blood and bile, the extent of gallbladder inflammation (eosinophil, neutrophil, lymphocyte), and the presence of adenomyomatosis and cholesterolosis in GB and stone composition (calcium bilirubinate, cholesterol, mixed, and protein) were all reviewed.

### 2.2. Definition

AC severity was assessed according to the TGs [2,6]. Grade I (mild) was defined as AC in a healthy patient with no organ dysfunction and mild inflammatory changes in the GB. Grade II (moderate) was defined as AC associated with any one of the following conditions: (1) elevated WBC count (>18,000/mm^3^); (2) palpable tender mass in the right upper abdominal quadrant; (3) duration of symptoms > 72 h; and (4) marked local inflammation (gangrenous cholecystitis, pericholecystic abscess, hepatic abscess, biliary peritonitis, emphysematous cholecystitis). Grade III (severe) was defined as AC associated with dysfunction of any one of the following organs/systems: (1) hypotension requiring treatment with dopamine > 5 ug/kg per min, or any dose of norepinephrine; (2) decreased level of consciousness; (3) PaO_2_/FiO_2_ ratio < 300; (4) oliguria, creatinine >2.0 mg/dL; (5) PT-INR > 1.5, and (6) platelet count < 100,000/mm^3^. Surgically resected gallbladder samples were obtained for regular histopathological analysis, which were fixed in neutral buffered formalin and inserted into paraffin. The samples were stained with hematoxylin and eosin. The number of inflammatory cells in the specimens was quantified separately. The extent of GB inflammation was defined by the number of eosinophils, neutrophils, and lymphocytes per high power field (HPF) in any layer [7,8].

### 2.3. Study Outcome

We compared demographic data (age, sex), laboratory findings (CRP, WBC, PLT, Cr, Alb, Bilirubin, AST, ALT, amylase, lipase, PT-INR), the microbiologic culture of blood and bile, the extent of gallbladder inflammation (eosinophil, neutrophil, lymphocyte), the presence of adenomyomatosis and cholesterolosis in GB, and stone composition (calcium bilirubinate, cholesterol, mixed, and protein) in accordance with the TGs.

### 2.4. Statistical Analysis

The chi-square test or Fisher’s exact test for categorical variables and Student’s t-test for continuous variables were used to compare the three groups. A *p*-value of <0.05 was considered statistically significant. All statistical analyses were performed by using SPSS software, version 18.0 (IBM, Armonk, NY, USA).

## 3. Results

A total of 217 patients were divided into three groups of increasing severity—Grade I (n = 146), Grade II (n = 51), and Grade III (n = 20)—in accordance with the TGs. The Grade III group contained significantly older patients than the Grade I or Grade II group (Grade I, 56.9 ± 13.9; Grade II, 64.3 ± 15.4; Grade III, 69.9 ± 9.9; *p*-value < 0.001). In the laboratory exam, the Grade III group showed significantly higher levels of CRP, WBC, creatinine, and bilirubin and significantly lower levels of PLT and albumin compared with the Grade I or Grade II group. However, there were no significant differences in terms of AST, ALT, and ALP. The detailed demographic and laboratory characteristics as stated by the TGs are summarized in Table 1.

Altogether, 2.3% (5/217) and 27.6% (60/217) of patients had microbiologic cultures of blood and bile, respectively. As the grade of severity increased, the rate of microbiological isolation significantly increased in both the blood (Grade I, 0% [0/146]; Grade II, 2.0% [1/51]; Grade III, 20% [4/20]; *p*-value < 0.001) and bile (Grade I, 19.9% [29/146]; Grade II, 33.3% [17/51]; Grade III, 70% [14/20]; *p*-value < 0.001). The microbiologic culture rates of each group are shown in Table 2 and the bacteria identified in the blood and bile did not vary with grading.

The extent of GB inflammation from the histopathological GB specimens taken after cholecystectomy is summarized in Table 3. There were no significant differences in terms of eosinophils, neutrophils, lymphocytes, adenomyomatosis, or cholesterolosis. GB stone composition was observed in Grade I (84.2%, 123/146), Grade II (78.4%; 40/51), and Grade III (75%; 15/20) cases (*p*-value = 0.01). Stones were generally composed of calcium bilirubinate in Grade II and III cases, and cholesterol in Grade I cases. Stone composition by TGs grading is shown in Table 4.

## 4. Discussion

In this study, we compared the extent of GB inflammation on the histopathological and laboratory findings of patients who had undergone cholecystectomy for AC with the severity of their cases as stated by the TGs. The patients in Grade III were significantly older than the patients in the Grade I or Grade II groups, and had significantly higher levels of CRP, WBC, creatinine, and bilirubin. The rate of microbiological isolation of blood and bile increased significantly with TG severity. However, there were no significant differences in terms of histologic findings. Based on current findings, the severity of AC according to the TGs does not correlate with the extent of GB inflammation on histopathology and laboratory findings. However, the microbiological isolation in blood and bile increased proportionally with the TG grade. AC begins initially as a chemical inflammation but is regularly complicated by bacterial invasion from the gut. *Escherichia coli*, *Klebsiella*, and *Streptococcus faecalis* predominate among the aerobic bacteria found, whereas *Bacteroides fragilis* and *Clostridia* are commonly encountered anaerobes. Mixed infections are prevalent. *Bactibilia* occurs in at least 60% of early-stage AC cases and is particularly prevalent among the elderly [9].

AC is defined as an inflammation of the GB, which is usually caused by gallstones obstructing the GB neck or cystic duct [10]. Although it is most often attributed to gallstones, many factors such as ischemia, motility disorders, and bacterial infection can be involved. AC mirrors this pathological condition and can be treated conservatively or by temporary drainage in the acute phase. However, a GB that has been damaged by inflammation can only be treated by surgical resection [3].

Although AC is common, there were no criteria for diagnosis or severity assessment and no established principles of clinical practice before 2007 [11].

The TGs were established in 2007 [10], and were revised and updated in 2013 [6] and 2018 [2]. The TGs are used worldwide in the diagnosis and treatment of AC in both clinical practice and research. The TGs classify AC into three severity grades using clinical symptoms, physical examination, blood tests, and findings from diagnostic imaging modalities. Grade I describes mild GB inflammation and Grade II describes moderate GB inflammation, while Grade III describes severe GB inflammation and is often associated with organ failure [6]. The TGs are clinically useful for classifying the severity of AC disease and for presenting a treatment algorithm.

Although acute cholecystitis usually has a good prognosis with timely diagnosis and management, complications associated with cholecystitis can be a significant challenge in clinical practice. Complications of acute cholecystitis can be caused by secondary bacterial infection or increased mural ischemia. Typical subtypes of complicated cholecystitis include hemorrhagic, gangrenous, and emphysematous cholecystitis, as well as gallbladder perforation. Complicated cholecystitis may result in significant morbidity and mortality. Therefore, early diagnosis and recognition are vital for management and treatment plans, including early cholecystectomy or percutaneous gallbladder drainage.

Laparoscopic cholecystectomy is the treatment of choice for AC [12,13]. The surgical risk of the cholecystectomy itself is relatively low, but if the patient is unstable, percutaneous cholecystostomy can be performed, or surgery can be delayed [14]. For low-risk patients, early surgery is recommended, as a cholecystectomy can be safely performed within 48–72 h of disease onset, during the congestive or edematous phase. Surgical risk should be assessed by the extent of the general condition of the patient. For patients with high surgical risk and severe GB inflammation, GB drainage can be performed as a temporary, and potentially life-saving, treatment. If the patient’s condition subsequently improves, elective surgery can be performed. Similarly to early surgery, GB drainage can also prevent serious complications of AC.

This study has several inherent limitations. First, this study aimed to determine whether the TG grading of AC is consistent with the extent of GB inflammation based on histopathological findings and microbiological isolation from blood and bile. A total of 217 patients were divided into three groups with increasing severity, i.e., Grade I (n = 146), Grade II (n = 51), and Grade III (n = 20), in accordance with the TGs. The imbalanced size of the three groups was inevitable due to the retrospective design of this study. Second, the TGs are generally based on the clinical or inflammatory conditions of patients, which serve as a guide for treatment. Therefore, although the surgical condition is important, this may be less relevant clinically. Third, in Western countries, carbapenem-resistant *Enterobacteriaceae* infections of the bile are currently a major problem. However, carbapenem-resistant *Enterobacteriaceae* infections are uncommon in South Korea, and such infections were not identified in this study.

In conclusion, despite microbiological isolation in blood and bile increasing with TG grading, the TGs do not reflect the histopathological severity of GB inflammation. This study suggests that the TGs reflect clinical severity but not pathologic severity. Calcium bilirubinate levels in gallstones were significantly higher in Grade III patients. Calcium bilirubinate is thought to be associated with infection, but further study is needed to confirm this issue.

## Figures and Tables

**Table 1 jpm-13-01335-t001:** Baseline characteristics.

	Grade I (n = 146)	Grade II (n = 51)	Grade III (n = 20)	*p*-Value
Age, years, mean ± SD	56.9 ± 13.9	64.3 ± 15.4 *	69.9 ± 9.9 *	<0.001
Sex, male:female	55:91	25:26	17:3 *	<0.001
CRP, mg/dL	2.95 (1.13–12.53)	12.50 (1.90–128.90) *	230.10 (97.88–289.88) *†	<0.001
WBC, /mm^3^	7495 (5990.0–9927.5)	14,090.0 (9800.0–18,240.0) *	14,865.0 (10,050.0–19,475.0) *	<0.001
PLT, /mm^3^	227,000 (190,000–268,000)	236,000 (209,000–272,000)	131,500 (88,000–213,750) *†	<0.001
Cr, mg/dL	0.72 (0.62–0.84)	0.81 (0.69–0.95) *	0.99 (0.84–1.24) *	<0.001
Albumin, mg/dL	4.30 (4.10–4.50)	4.20 (3.80–4.40) *	3.70 (3.35–3.95) *†	<0.001
Bilirubin, mg/dL	0.70 (0.50–1.20)	0.80 (0.50–1.30)	1.60 (0.80–3.10) *†	<0.001
AST, IU/L	29.0 (21.0–52.5)	30.0 (21.0–42.0)	89.0 (29.3–151.8) *†	0.032
ALT, IU/L	27.5 (15.0–56.5)	24.0 (16.0–45.0)	54.5 (18.5–93.8)	0.257
ALP, IU/L	77.0 (63.8–98.5)	79.0 (65.0–113.0)	81.0 (71.3–145.8)	0.0220
Amylase, IU/L	50.0 (40.8–61.0)	31.0 (25.0–53.0) *	3.0 (21.3–35.0) *	<0.001
Lipase, IU/L	18.0 (12.0–26.0)	12.0 (7.0–22.0) *	12.5 (6.5–16.0) *	<0.001
PT-INR	1.06 (1.01–1.10)	1.14 (1.07–1.22) *	1.28 (1.18–1.48) *†	<0.001

* *p* < 0.05 compared with Grade I. † *p* < 0.05 compared with Grade II. Data were presented as mean ± standard deviation for normally distributed data, and median (Q1 to Q3) for non-normally distributed data. CRP, C-reactive protein; WBC, white blood cell; PLT, platelet; Cr, creatinine; AST, aspartate aminotransferase; ALT, alanine aminotransferase; PT, prothrombin time; INR, international normalized ratio.

**Table 2 jpm-13-01335-t002:** Microbiologic cultures according to the TGs.

	Grade I (n = 146)	Grade II (n = 51)	Grade III (n = 20)	Total (n = 217)	*p*-Value
Blood	0 (0/146)	* 2.0% (1/51)	** 20% (4/20)	2.3 (5/217)	<0.001
Bile	† 19.9% (29/146)	†† 33.3% (17/51)	††† 70% (14/20)	27.6% (60/217)	<0.001

* Escherichia coli (n = 1). ** Escherichia coli (n = 4). † Escherichia coli (n = 7), Klebsiella pneumoniae (n = 7), Enterococcus faecium (n = 4), Staphylococcus aureus (n = 2), Enterobacter aerogenes (n = 1), Klebsiella oxytoca (n = 1), Achromobacter xylosoxidans (n = 1), Enterococcus faecakis (n = 1), Enterococcus hirae (n = 1), Citrobacter freundii (n = 1). †† Escherichia coli (n = 8), Klebsiella pneumoniae (n = 5), Enterobacter aerogenes (n = 2), Klebsiella oxytoca 1 (n = 1), Enterobacter cloacae (n = 1). ††† Escherichia coli (n = 5), Klebsiella pneumoniae (n = 3), Enterobacter aerogenes (n = 1), Enterobacter cloacae (n = 1), Staphylococcus aureus (n = 1), Enterococcus hirae (n = 1), Citrobacter freundii (n = 1), Enterococcus gallinarum (n = 1).

**Table 3 jpm-13-01335-t003:** Pathologic findings of gallbladder according to the TGs.

	Grade I (n = 146)	Grade II (n = 51)	Grade III (n = 20)	*p*-Value
Eosinophils	1.0 (0.0–1.0)	1.0 (0.0–2.0)	0.0 (0.0–1.0)	0.211
Neutrophils	0.0 (0.0–1.0)	0.0 (0.0–2.0)	0.5 (0.0–2.0)	<0.001
Lymphocytes	1.0 (1.0–2.0)	2.0 (1.0–2.0)	1.0 (1.0–2.0)	0.522
Adenomyoma	0.0 (0.0–0.0)	0.0 (0.0–0.0)	0.0 (0.0–0.0)	0.270
Cholesterolosis	13 (9.1%)	1 (2.0%)	0 (0.0%)	0.101

**Table 4 jpm-13-01335-t004:** Stone composition according to the TGs.

	Grade I (n = 123)	Grade II (n = 40)	Grade III (n = 15)	*p*-Value
	84.2% (123/146)	78.4% (40/51)	75% (15/20)	0.010
Calcium bilirubinate	40.7% (50/123)	70.0% (28/40)	80% (12/15)	
Cholesterol	48.0% (59/123)	22.5% (9/40)	13.3% (2/15)	
Mixed	9.8% (12/123)	7.5% (3/40)	6.7% (1/15)	
Protein stones	1.6% (2/123)	0	0	

## Data Availability

The data used to support the findings of this study are available from the corresponding author upon request.

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
