# Peer review of "Relationship between the Tokyo Guidelines and Pathological Severity in Acute Cholecystitis"

_jpm, 2023, doi:10.3390/jpm13091335_

Round 1
Reviewer 1 Report
This retrospective study implied a thorough work to retrieve a high volume of biological and microbiological data with a serious statistical analysis. However, being a retrospective study with quite imbalanced number of patients included in the three studied groups, is an important drawback of the paper.
This a thorough research correlated with a high incidence surgical situation. However it seems to be of less clinical impact, since TG is mostly correlated with clinical and inflammatory patients condition and this is what practically guides the treatment.
Author Response
This retrospective study implied a thorough work to retrieve a high volume of biological and microbiological data with a serious statistical analysis.
Point 1: Being a retrospective study with quite imbalanced number of patients included in the three studied groups, is an important drawback of the paper.
Response 1: Authors appreciate reviewer’s helpful comment. This study aimed to determine whether the TGs grading of AC is consistent with the extent of GB inflammation in the histopathological findings and microbiological isolation in blood and bile. A total of 217 patients were divided into three groups of increasing severity Grade I (n=146), Grade II (n=51), and Grade III (n=20) in accordance with the TGs. The imbalanced number of three studies groups was inveitable due to retrospective design of this study. This can be a limitation. Regarding this point, we have added a paragraph as limitation of our study in “Discussion-limitation” section as follows:
“This study aimed to determine whether the TGs grading of AC is consistent with the extent of GB inflammation in the histopathological findings and microbiological isolation in blood and bile. A total of 217 patients were divided into three groups of increasing severity Grade I (n=146), Grade II (n=51), and Grade III (n=20) in accordance with the TGs. The imbalanced number of three studies groups was inveitable due to retrospective design of this study”
Point 2: This a thorough research correlated with a high incidence surgical situation. However it seems to be of less clinical impact, since TG is mostly correlated with clinical and inflammatory patients condition and this is what practically guides the treatment.
Response 2: Authors agree with reviewer’s opinion. Weak clinical impact is a major limitation of this study, and we have added a paragraph as in limitation section as follows:
“Since TG is mostly correlated with clinical and inflammatory patients condition, and this is what practically guides the treatment.Therefore, Although, thorough research correlated with a high incidence surgical situation, it can be of less clinical impact.”
Reviewer 2 Report
It is a valuable study, very well carried out, in which it is found that the degree of clinical severity of acute cholecystitis does not correlates to a similar histopathological severity.
In Western countries, carbapenem-resistant Enterobacteriaceae infections of the bile are now a serious problem. In the study, there is no mention of any such infection, I suppose in Korea these types of infections are uncommon. Maybe some comment could be interesting.
Some minor corrections are listed below.
Some words are missing in the following sections, the underlined word must be included:
Abstract, Results, last line
However, there were no significant differences in the extent of GB inflammation between grades
Intoduction
depending on the patient’s surgical condition, the severity of GB inflammation,
Methods, Defiinitions
The extent of GB inflammation was defined by the presence of the number
of eosinophils, neutrophils, lymphocyte count was defined by the number of cells per
high power field (HPF) in any layer
Only minor editing required
Author Response
It is a valuable study, very well carried out, in which it is found that the degree of clinical severity of acute cholecystitis does not correlates to a similar histopathological severity.
Point 1: In Western countries, carbapenem-resistant Enterobacteriaceae infections of the bile are now a serious problem. In the study, there is no mention of any such infection, I suppose in Korea these types of infections are uncommon. Maybe some comment could be interesting.
Response 1: Authors appreciate reviewer’s excellent comment. Carbapenem-resistant Enterobacteriaceae infections of the bile also can be a serious clinical problem in South Korea. However, carbapenem-resistant Enterobacteriaceae infections is uncommon, and such infections were not indentified in this study. Regarding this point, we have added a paragraph as limitation of our study in “Discussion-limitation” section as follows:
“In Western countries, carbapenem-resistant Enterobacteriaceae infections of the bile are now a serious problem. However, carbapenem-resistant Enterobacteriaceae infections is uncommon, and such infections were not indentified in this study.”
Point 2: Some minor corrections are listed below. Some words are missing in the following sections, the underlined word must be included:
Response 2: Authors appreciate reviewer’s helpful comment. Indicated missing words were corrected in the manuscript.
Abstract, Results, last line: However, there were no significant differences in the extent of GB inflammation between grades
Intoduction: depending on the patient’s surgical condition, the severity of GB inflammation,
Methods, Defiinitions: The extent of GB inflammation was defined by the presence of the numberof eosinophils, neutrophils, lymphocyte count was defined by the number of cells perhigh power field (HPF) in any layer
Round 2
Reviewer 1 Report
The main point of your paper is the originality since there seems no other paper was published with this topic. Unfortunately the rest of the comments are still of actuality despite some new paragraphs that underline them
Although providing an interesting hypothesis and with some explanatory new lines, unfortunately I still think that the study is of less quality than required for JPM. However the editorial committee might take into consideration to accept the paper due to the originality of the subject;in this situation I would strongly recommend a thorough english revision especially for the new added lines.
Author Response
The main point of your paper is the originality since there seems no other paper was published with this topic. Unfortunately the rest of the comments are still of actuality despite some new paragraphs that underline them Although providing an interesting hypothesis and with some explanatory new lines, unfortunately I still think that the study is of less quality than required for JPM. However the editorial committee might take into consideration to accept the paper due to the originality of the subject;in this situation I would strongly recommend a thorough english revision especially for the new added lines.
Response 2: Authors appreciate reviewer’s comment. English revision for the new added lines were performed as follows:
Introduction section
From: “Over the years, it has been argued that AC can be primarily conservatively treated and subsequently delayed selective cholecystectomy. Evidence has been collected in recent decades to show that early cholecystectomy for AC during acute hospitalization is safe and cost-effective. Recently, data have emerged in centers with a lot of experience in laparoscopic surgery to show that early laparoscopic cholecystectomy is an attractive, feasible and safe treatment strategy.”
To: “Over the years, it has been claimed that AC can be conservatively treated in most cases, consequently delaying selective cholecystectomy. However, evidence accumulated in recent years has shown that early cholecystectomy for AC during acute hospitalization is safe and cost-effective. Specifically, data collected from centers with extensive experience in laparoscopic surgery suggest that early laparoscopic cholecystectomy may be an attractive, feasible, and safe treatment strategy.”
Method-definition section
From: “The surgical resected gallbladder samples were taken for regular histopathological analysis, fixed in neutral buffered formalin, and inserted into paraffin. The area was stained with hematoxylin and eosin. The amount of inflammatory cells was scored separately on the specimens. The extent of GB inflammation was defined by the presence of the number of eosinophils, neutrophils, lymphocyte count was defined by the number of cells per high power field (HPF) in any layer7, 8”
To: “Surgically resected gallbladder samples were obtained for regular histopathological analysis, which were fixed in neutral buffered formalin and inserted into paraffin. The samples were stained with hematoxylin and eosin. The number of inflammatory cells in the specimens was quantified separately. The extent of GB inflammation was defined by the number of eosinophils, neutrophils, and lymphocytes per high power field (HPF) in any layer.7, 8”
Discussion section
From: “Although, acute cholecystitis usually has as good prognosis with timely diagnosis and management, complications associated with cholecystitis can be a significant challenge in clinical practice. Complications of acute cholecystitis can be caused by secondary bacterial infection or increased mural ischemia. Typical subtypes of complicated cholecystitis include hemorrhagic, gangrene, emphysematous cholecystitis, and gallbladder perforation. Complicated cholecystitis may cause significant morbidity and mortality, Therefore, early diagnosis and recognition play a important role in the management and treatment plans including early cholecyetctomy or percutaneous gallbladder drainage.”
To: “Although acute cholecystitis usually has a good prognosis with timely diagnosis and management, complications associated with cholecystitis can be a significant challenge in clinical practice. Complications of acute cholecystitis can be caused by secondary bacterial infection or increased mural ischemia. Typical subtypes of complicated cholecystitis include hemorrhagic, gangrene, emphysematous cholecystitis as well as gallbladder perforation. Complicated cholecystitis may result in significant morbidity and mortality. Therefore, early diagnosis and recognition are vital for management and treatment plans including early cholecystectomy or percutaneous gallbladder drainage.”
From: “This study has several notable inherent limitations. First, This study aimed to determine whether the TGs grading of AC is consistent with the extent of GB inflammation in the histopathological findings and microbiological isolation in blood and bile. A total of 217 patients were divided into three groups of increasing severity Grade I (n=146), Grade II (n=51), and Grade III (n=20) in accordance with the TGs. The imbalanced number of three studies groups was inveitable due to retrospective design of this study. Second, since TG is mostly correlated with clinical and inflammatory patients condition, and this is what practically guides the treatment.Therefore, Although, thorough research correlated with a high incidence surgical situation, it can be of less clinical impact. Third, in Western countries, carbapenem-resistant Enterobacteriaceae infections of the bile are now a serious problem. However, carbapenem-resistant Enterobacteriaceae infections is uncommon, and such infections were not indentified in this study.”
To: “This study has several inherent limitations. First, this study aimed to determine whether the TGs grading of AC is consistent with the extent of GB inflammation based on histopathological findings and microbiological isolation from blood and bile. A total of 217 patients were divided into three groups with increasing severity, i.e., Grade I (n=146), Grade II (n=51), and Grade III (n=20), in accordance with the TGs. The imbalanced size of the three groups was inevitable due to the retrospective design of this study. Second, the TGs are generally based on the clinical or inflammatory conditions of patients, which serve as a guide for treatment. Therefore, although the surgical condition is important, this may be less relevant clinically. Third, in Western countries, carbapenem-resistant Enterobacteriaceae infections of the bile are currently a major problem. However, carbapenem-resistant Enterobacteriaceae infections are uncommon in South Korea, and such infections were not identified in this study.”